# Planning Perspectives on Rural Connected, Autonomous and Electric Vehicle Implementation

Joseph G. Walters [1,*], Stuart Marsh [1] and Lucelia Rodrigues [2]

1   Nottingham Geospatial Institute, University of Nottingham, Triumph Road, Nottingham NG7 2TU, UK; stuart.marsh@nottingham.ac.uk
2   Department of Architecture and Built Environment, University of Nottingham, University Park, Nottingham NG7 2RD, UK; lucelia.rodrigues@nottingham.ac.uk
*   Correspondence: joseph.walters@nottingham.ac.uk or joegewa@gmail.com

**Abstract:** Connected, autonomous and electric vehicles (CAEV) are a powerful combined transport technology looking to disrupt the automotive sector and drive the transition to safe, accessible, clean and sustainable transport systems. The trialling of private, public and shared CAEV technologies is occurring in cities around the world; however, historically isolated and transport-poor rural communities may have the most to gain from CAEV implementation. Despite the accessibility and transport safety needs of rural communities, rural CAEV trials are few in the UK. Therefore, this paper investigates the hypothesis that the lack of rural implementation research and trials means that rural transport planners are ill-informed and uncertain of both the potential of CAEVs and their implementation requirements to meet rural community transport needs. This investigation consists of consultations with UK-based transport planning professionals to establish their perspectives on CAEV technologies and their rural implementation potential. The findings show that 96% of transport planners lack sufficient understanding of CAEV technology and its implementation challenges. However, the findings also highlight a willingness, given the opportunity, for transport planners to engage with CAEV technologies and apply them to specifically rural scenarios.

**Keywords:** connected vehicles; autonomous vehicles; electric vehicles; rural communities; sustainable transport; accessibility; transport planning; intelligent transport systems

## 1. Introduction

### 1.1. Rural Transport Challenges

Agricultural activity has defined rural characteristics for many years resulting in a lack of development within communities and an ignorance of rural social issues outside them [1]. However, the economic structures of rural working communities are becoming more diverse, along with social and environmental structures [2,3], with continuing trends of counter-urbanisation confirming the ongoing appeal of rural community life [4–7]. Despite this, rural communities continue to have higher levels of poverty, social exclusion and inequality compared with urban communities, which can be directly linked to physical isolation and a lack of accessibility due to scattered and peripheral rural characteristics [2,8–13]. Improving accessibility through the development of transport solutions is therefore fundamental to improving the rural socio-economic systems they serve [8,14]; however, complex rural mobility dynamics can make the estimation of movement and delivery of effective transport solutions challenging [15,16]. Transport enables mobility to ensure access to markets and resources, which promotes economic and social growth, improves quality of life, reduces social exclusion and increases accessibility to public services and opportunities [1,15,17].

Whilst congestion, traffic accidents and mobility gaps all create transport challenges, the sustainability and decarbonisation of transport have become additional central issues [18]. For transport to be sustainable, mobility solutions must be equitable, efficient,

safe and climate responsive [19]. Integrated private, public and active zero-carbon transport systems are needed to meet the environmental, social and economic needs of rural communities [17,19]. Reducing emissions is critical for the sustainable development of the global transport sector. Since their inception, private cars have continually provided people with unprecedented levels of mobility and accessibility; however, increased car ownership has had negative impacts on the environment that now need to be mitigated [14,15], particularly in rural communities that are highly car-dependent [8]. The development and implementation of alternative fuelled vehicles such as battery electric vehicles (BEV) or hydrogen fuel cell electric vehicles (FCEV), which produce zero emissions on the road, is one action that is already making ground.

Safety is also fundamental to a sustainable transport system [20] which must avoid fatalities, injuries and crashes relating to transport. Ninety-seven percent of transport deaths worldwide are road transport related and account for 93% of costs [19]. In the UK, data from the Department for Transport (DfT) indicate that consistently over 50% of traffic fatalities occur on rural roads annually [21]. This is in part due to the irregular, winding and narrow nature of rural roads, in contrast to the typically uniform and predictable nature of urban roads and motorways [22]. In addition, human error is the causal factor for over 90% of traffic accidents [23,24].

Universal access requires transport systems to be inclusive and a continued lack of access to services and technology keeps rural communities isolated from their urban counterparts [2,13,25]. The challenge of providing access and connectivity to rural communities is enhanced by the combination of poor physical transport services as well as digital exclusion, where the contribution of digital technologies to more effective public transport services [10,19] is blocked by a lack of connectivity.

Advancing transport technologies can address many of these rural social, economic and environmental challenges, and help close the development gap between rural and urban communities [11]. Connected, Autonomous and Electric Vehicles (CAEV) are one such technology, or more accurately a combination of technologies. It is important that the potential benefits of CAEVs will not only be experienced by those living in urban areas, especially as it is rural communities that are isolated, lack accessibility, who are at most risk of severe traffic accidents, and who lack digital connectivity. Rural communities have a significant amount to gain from CAEV technologies and hence rural priorities and considerations should be central to future CAEV developments. However, despite this potential, rural CAEV trials and implementation are lacking in the UK.

### 1.2. Hypothesis and Structure

This paper hypothesises that the lack of rural implementation research and trials means that transport planners are ill-informed and uncertain of both the potential of CAEVs and their implementation requirements to meet rural community transport needs. Therefore, the purpose of this paper is to explore this hypothesis through consultations with UK-based rural transport planning professionals. As such, this paper contributes a qualitative investigation into the state and readiness of the transport planning industry regarding rural CAEV implementation.

It is rural transport planners that will be responsible for practically implementing the infrastructural and technological changes needed to support rural CAEV implementation. They can do this by designing and developing policies, plans and strategies considering economic, social and environmental factors and working with governments, communities and other built-environment disciplines to implement solutions. Transport planners should understand the current and future transport systems that specifically cater for the communities which they serve, and are involved in the life-cycle of sustainable transport solutions and decision making processes [26]. Whilst this paper directly refers to rural transport planners, it recognises that transport planning is a diverse discipline and that planners may work in a variety of roles that may not always be specifically rural or transport related.

For context, this paper initially identifies some of the requirements and improvements that are needed in order to deliver CAEVs and their benefits to rural roads for rural communities. However, the practicalities of such improvements are not directly considered. The hypothesis is then tested using a methodology which describes the process of engaging with transport planning professionals to establish their perspectives on the issue of rural CAEV implementation. The results from these consultations are then discussed and conclusions generated. Finally, the authors make recommendations for further study on this research topic.

In addition to testing the hypothesis, this paper makes several research contributions:

1. This paper addresses the unique challenges facing the rural implementation of CAEV technologies, hard and soft supporting infrastructure, and stakeholder engagement and understanding explicitly, for the first time;
2. This paper demonstrates active engagement with transport planners to gather and contribute quantitative and qualitative evidence highlighting the extent of the gap in understanding and knowledge of CAEVs, their technologies and their rural transport potential;
3. Finally, this paper contributes a base from which further studies can build and accelerate transport planner understanding of rural CAEV implementation in the UK and globally.

## 2. Connected, Autonomous and Electric Vehicles

Connected, Autonomous and Electric Vehicles (CAEVs) bring together several transport technologies to provide a solution capable of wireless connectivity and autonomous driving functions, which are powered by an electric-based power source. The combination of these technologies is predicted to positively transform mobility services at reduced costs [27]. Whilst CAEVs have broad applications across a variety of sectors including private and public transport, rail, aerial, marine, agriculture and working in hazardous environments, this paper focuses primarily on road-based public, private and shared motorised vehicles.

As CAEVs are a developing technology, there are varying levels of connectivity, autonomy and electric-power options available and emerging. For road-based vehicles, there is an international standard defined by the Society of Automotive Engineers (SAE) on the terms related to driving automation systems which define six levels of autonomy from Level 0: No Driving Automation to Level 5: Full Driving Automation [28]. Each level details the autonomous functions of a vehicle and defines the level of human interaction with Artificial Intelligence (AI), if any, required to operate the vehicle [20,29]. Effective real-time positioning is required for CAEV autonomy and can be achieved through a combination of on-board positioning, satellite and mapping techniques. On-board positioning is required for relative positioning in the surrounding environment and utilises a range of sensor technology including radar, LiDAR (light detection and ranging) and cameras, to continuously monitor the environment to prevent collisions and maintain safe autonomy [30]. Satellite positioning meanwhile is required to determine a CAEVs global position and location on roads by using the Global Navigation Satellite System (GNSS) [31]. In the UK, network real-time kinematic positioning (NRTK) can improve navigation flexibility, minimise errors and improve positioning range [32,33]. Mapping meanwhile, made available through connectivity, can be used assist these positioning methods. Wireless technologies provide the connectivity required to enhance the positioning capabilities of the CAEV. Further, vehicle-to-everything (V2X) communication technologies, which enable an individual vehicle to connect to other vehicles, infrastructure and other internet-of-things (IoT) objects, promote a cooperative driving experience to collectively enhance safety and improve fuel efficiency [34]. The rapid adoption of private and public BEVs in the UK supports the findings that connected and autonomous vehicles will rely on electric battery power [35,36]. This is preferable to internal combustion engine (ICE) powered vehicles for seamless connected and autonomous integration with vehicle systems

and software [37]. Over ICEs, BEVs benefit from lower operating costs, greater efficiencies and minimal environmental impact [36].

CAEVs have the potential to provide effective and sustainable transport solutions that are capable of addressing many of the rural transport problems described in this paper. For example, replacing, or supplementing, traditional public transport with alternative options such as demand responsive transport (DRT) can improve rural accessibility in a sustainable way [8,38,39]. DRT can make use of many of the features of CAEVs including connectivity required to request the vehicle via a connected device such as a smartphone, and autonomy to locate the user and calculate efficient route options particularly in shared transport scenarios. Mobility as a Service (MaaS) looks to support DRT though single travel management platforms that digitally unify the transport service process [40] and requires open data sharing and reliable wireless communication infrastructure [41]. Digital connectivity is therefore a crucial component for CAEV operation, positioning and safety; however, current networks mean that digital infrastructure in the UK presently best serves urban communities and is not equitable across rural communities [13,42]. As CAEV technology rapidly improves, the digital infrastructure required to support it on UK roads needs to be developed quickly. This is particularly vital in currently severely lacking rural areas and so fully connecting rural UK areas is arguably more important than improving the connectivity speeds of those who already have it [21]. Strong digital connectivity is key to strengthening rural economies, and is critical to future transport systems [11,19]. Although the digital divide between rural and urban communities is itself a barrier to rural CAEV implementation, the need to implement sustainable transport solutions may accelerate the improvements to connectivity in rural communities and on rural roads, as it has done in the context smart cities [43]. Therefore, the implementation of CAEVs in rural areas provides an opportunity for improved rural digital connectivity.

To support the integration of CAEVs in rural areas, the mobility behaviours of rural populations, which are typically observed on a relatively small regional scale [16], would need to be understood. Such mathematical models predicting dynamic human mobility include the gravity model [44] and the resulting radiation model which successfully predicts a wide range of mobility patterns including inter-regional movement inclusive of rural-orientated commutes [45]. An understanding of these models and how they can support CAEV implementation would aid the rollout of rural DRT and MaaS services. Further, as CAEVs disrupt traditional driving and mobility habits [20], the communication and GNSS data generated from CAEV movement post-implementation have the potential to contribute to future mobility and spatial studies [15], and the improvement of CAEV services. Initially, however, the spatial resolution of data used to predict rural movements is likely to be less reliable compared to urban areas due to a lack of connected infrastructure, for example, cell towers [15]. Such mobility and spatial models as the Unified Mobility Estimation Model (UMEM) aim to reduce reliance on data requirements for such reasons, focusing on mobility estimation using existing population, points of interest and road network data [16]. As working methods continue to evolve post-COVID and integrate hybrid and home-based working, such mobility methods as those above may require further revisions to support CAEV implementation, particularly in the rural context with sparse and small population centres. In addition, changing mobility habits due to the pandemic offer opportunities for CAEVs to deliver contactless services [46,47] with the additional rural-specific advantage of potentially protecting isolated communities from transmission. Such investigations, however, are beyond the scope of this paper.

The efficiency benefits that could be achieved through autonomous and connected DRT and MaaS look to reduce congestion and journey times for users. Such congestion improvements can be achieved with wide-spread levels of adoption of highly-automated vehicles [20,48]. To improve road safety CAEVs are designed to either completely eliminate human error [49] or enhance driver performance with automated functions such as brake assist [50]. Complete vehicle automation is expected to significantly improve road safety [20,24] with connectivity used to enhance safety through information sharing [35,36]. The use of batteries as a power

source for CAEVs will reduce noise and particulate pollution at local community levels and reduce the associated transport health risks of local pollution. At a wider scale, CAEV reduce transport contributions to national greenhouse gas (GHG) emissions [35]. With UK energy supplies increasingly provided by renewable energy sources such as wind energy, the environmental impacts of CAEVs will continue to decrease. Economically, CAEVs have the potential to support local communities through transport cost savings from improved ease of travel, reduced accident costs, improved productivity and increased trade. Once successfully implemented, the provision of CAEV DRT and MaaS services will have substantial economic benefits [35,38,41] and there is also potential for job-creation including the maintenance, monitoring and operation of individual vehicles or fleets [51].

For CAEVs to be successfully and sustainably implemented in rural communities, they need to be accessible and safe, positively impact local and wider societies and economies, and not compromise the health of the environment. Widley cited initial barriers to the implementation of CAEVs include the effectiveness of the technology, high technological costs, experimental and untested technology, integration with existing road traffic, regulatory challenges and societal acceptance [13,20,49,51–57]. The issue of social acceptance is one of the most researched CAEV challenges. There are numerous factors contributing to the level of user acceptance regarding CAEVs including: the complexity of integrated human and automated driving [50,58,59]; the historic and perceived capabilities of CAEV technology [52,60]; individual and societal ethical perspectives and the perceived ethical responsibility of AI systems [61–63]; cyber-security concerns [63,64]; and willingness used shared automated transport [63,65]. Ultimately, the perceived implications of personal safety and security govern CAEV acceptance [63,66], resulting in a priori reluctance towards CAEVs [63]. Research surveys reveal the consequences of this and return mixed public opinion regarding CAEV technologies, with generally negative perceptions regarding the use of fully autonomous vehicles [51,63,67]. Of course, levels of acceptance vary across demographic groups, with young males typically the least reluctant to use public autonomous transport [63]. Demographic information such as age, gender and education and employment factors can be used to establish potential CAEV acceptance levels [68]. Further, cultural and educational issues relating to people's ability to use the internet, and as a consequence, misunderstand CAEV technologies and services such as ride-hailing, can also be a barrier [65], particularly in the rural context. Such psychological barriers as CAEV acceptance are difficult to overcome without physically introducing the technologies themselves into markets, after which, acceptance typically follows [63]. However, public trust can be strengthened through rigorous testing and certification; delivering evidence of effective CAEV situational awareness; and continual transparency of accuracy, reliability and sensing quality [52].

Real-time vehicle positioning, dynamic connectivity, and dynamic mapping are the three key technologies required for the successful development of CAEV transport. These technologies must be reliable, accurate and continuously available if CAEVs are to be an effective transport solution [69] and will require appropriate supporting infrastructure. In rural areas, the provision of this infrastructure can be challenging. Wireless connectivity in terms of 4G (4th Generation) cellular signals and the consistent readability of roads are two of the main infrastructural challenges facing rural CAEV implementation [21,34]. In addition, despite the accuracy of network real-time kinematic (NRTK) satellite positioning [70], it lacks availability when line of sight is interrupted, which is of particular concern on unpredictable and poorer quality rural roads [21,33].

Despite the challenges referenced above, studies on CAEV implementation in rural communities and on rural roads are limited, with the majority of the literature focused on urban CAEV implementation and associated infrastructure challenges [13,21].

## 3. Methodology

CAEVs have the potential to disrupt and improve rural transport systems, infrastructure and services. However, based on the present urban focus identified, it is assumed

that CAEVs and their potential implementation are rarely considered in rural transport planning. As such, it is unclear to what extent the transport planning profession is aware of, understands the technology behind, and believes in the specifically rural benefits of CAEVs and their technologies. Therefore, this paper describes an elicitation exercise conducted to answer the following questions:

- To what extent is the implementation of rural CAEV technology currently a priority for rural transport planning professionals?
- To what extent do rural transport planners believe that CAEVs are an important factor for the consideration of future sustainable transport solutions for rural communities?
- To what extent do rural transport planners understand CAEV technologies and their infrastructural requirements in the rural context?

Through an elicitation selection process [71], an elicitation approach combining surveys and semi-structured conversational interviews [72] was developed. The advantages and disadvantages of both survey and interview methods had the potential to impact the quality of this study. Therefore, both methods were presented to potential participants. The target participants for elicitation were specifically rural transport planning professionals. However, due to the breadth and complexity of the research subject, other professionals with strong links to rural transport development and/or CAEV development and implementation were also targeted. Therefore, the non-probability method of purposive sampling was used. This method is used across research industries to specifically target individuals with knowledge in a certain area so that the data collected are meaningful to the aims of the research [73–75]. Despite the associated bias of the method, purposive sampling is efficient in that the selected individuals are assumed to have knowledge of the research subject and any individuals with no knowledge of the subject are filtered out prior to the elicitation. However, selected individuals, although assumed to be knowledgeable, may not necessarily be reliable [76]. To further filter non-knowledgeable and non-reliable participants, the participant information section of both survey and interview forms made clear to potential participants the types of respondents required for the study prior to elicitation. In the initial stages of both survey and interview elicitations, questions are asked regarding participant employment and job status to gauge their knowledge, experience, and expertise of the subject matter. All participants in either survey or interview elicitations gave their informed consent for inclusion before participation. The study was conducted in accordance with the Declaration of Helsinki [77], and the protocol was approved by the University of Nottingham Faulty of Engineering Ethics Committee through consultation with the project researchers and supervisors.

An online survey was developed to collect quantitative and short qualitative data. To begin constructing the question schedule, a list of topics and related questions was drafted, relevant to the research aims. This list was transferred into a spreadsheet that split the questions according to their level of required detail and their relevance to the research. An iterative review process took place to develop the questions before a clear question structure was developed with the most important and relevant questions identified and grouped [78]. From this, the survey was developed to be used to collect the fundamental knowledge required to complete the minimum research requirements of this study. The survey question schedule can be found in Appendix A. Complementary to the survey, semi-structured interviews were carried out with specific professionals with known expertise to support, and scrutinise, the survey findings and the initial findings from the literature review element of this research in high qualitative detail. Semi-structured interviews were conducted as they are a commonly used method to understand participant perspectives on specific topics [71,79–81]. Similarly to the survey question schedule, the interview question schedule was extracted from the same questions spreadsheet with semi-structured interview methods in mind to develop an interview schedule. The interview question schedule can be found in Appendix A. A further iterative approach to interview development was adopted over the interview period as and when issues were identified in practice. However, adopting a semi-structured interview style allowed participants to veer away from this

direct line of questioning and into their own areas of expertise where they were more comfortable in sharing their knowledge.

The survey and interviews produced both quantitative and qualitative data types. The quantitative data was analysed using Microsoft Excel and SPSS software and statistical methods. The qualitative data was analysed using NVIVO software, using a reflexive thematic analysis method to generate codes and identify themes whilst allowing for the researcher's individual and subjective engagement with and interpretation of the data [82,83]. The analysis was split into two areas, the first addressing the defined research questions and the second assessing the participants knowledge based on their job role, title and experience. The specifics of the latter are not shared in this paper in accordance with the ethics protocol, to protect participant anonymity.

One of the drivers for this study is the identification of both a lack of rural CAEV trials and a focus on the development of urban CAEV solutions, despite rural communities potentially having a lot to gain from developing CAEV technologies. To support this, 74% of the survey participants "agreed" or "strongly agreed" that urban transport planning takes priority over rural transport planning, with no respondents "strongly disagreeing". No distinct pattern emerged regarding the locations of the respondents and their responses to this question. This supports the perceived nation-wide urban–rural transport divide.

For clarity, this paper uses the rural definitions developed by [84,85] which are used by the UK Government and Office for National Statistics (ONS). Rural areas and regions of the UK are built up using rural Output Areas (OA) which are small geographical areas with populations of less than 10,000. In the case of a Local Authority District (LAD), a common geographical scale used amongst transport planners, a rural LAD is one where over 50% of the LAD is made up of rurally defined OAs. This definition was made available to both survey and interview participants.

## 4. Elicitation Results and Discussion

### 4.1. Summary of Survey Responses

In total, 23 survey responses from professionals in transport planning and planning-related disciplines (referred to as "survey participants") were analysed supported by five detailed interview elicitations (referred to as "interview participants" $\alpha$, $\beta$, $\gamma$, $\delta$ or $\varepsilon$) with selected experienced transport professionals. Whilst a relatively small sample, the participants collectively represented at least 7 of the 12 regions of the UK inclusive of Scotland, the North East, Yorkshire and the Humber, the East Midlands, the West Midlands, the East of England and the South West. Each of these seven regions features significant rural geography and represent a variety of rural communities. Despite this, a more comprehensive study involving a greater number and range of participants would be required to substantiate the findings of this paper. Such a study is referred to in the Further Work and Recommendations section of this paper.

### 4.2. Priority of CAEV Implementation

The survey participants were asked to rank a list of priority areas for rural transport. The results, including the rank and average score out of a maximum of ten, are shown in Table 1. The option score was calculated based on weighting each rank-an option scored 10 if it was ranked first and 1 if it was ranked last–and averaged over the number of participants.

The highest priority areas for rural transport were "improving accessibility" with 50% of respondents selecting this as their first priority. "Affordability" and "safety" also rank highly, with "safety" being a more common first choice but having a greater range of responses, with some ranking "safety" their lowest priority. "Communications infrastructure" is ranked low, with "automation" ranked significantly lowest. Fifty-seven percent of respondents ranked "automation" their lowest priority. Whilst the other priority areas could be seen as more traditional in their relation to transport methods and practice, these lowest ranked elements relate directly to emerging transport technologies, specifically

CAEVs. "Communications infrastructure" may score more highly than "automation" due to the lack of general communications infrastructure in rural areas, but it remains low due to the current disassociation between communication and transportation infrastructure.

**Table 1.** Ranked priority areas for rural transport with weighted scores.

| Option | Rank | Score |
|---|---|---|
| Accessibility | 1 | 9.1 |
| Affordability | 2 | 7.3 |
| Public transport services | 3 | 6.2 |
| Safety | 4 | 5.8 |
| Quality of infrastructure | 5 | 5.2 |
| Sustainability | 6 | 5.1 |
| Maintenance of infrastructure | 7 | 4.7 |
| Environment quality | 8 | 4.7 |
| Communications infrastructure | 9 | 4.1 |
| Automation | 10 | 2.7 |

Considering new and developing transport technologies, just 12% of participants "agreed" that future transport systems and technologies were considered when planning rural transport systems and infrastructure. 50% "disagreed" that future transport systems were considered whereas 38% "neither agreed nor disagreed", suggesting that they are uncertain whether this consideration takes place. Alternatively, they may see the consideration of future transport technologies in some cases, but not in all cases. On average, there was a mild disagreement to the question amongst respondents.

With CAEVs, there is no "labour directly involved in operating the vehicles ... so automation gives you the opportunity to have much larger fleets of much smaller vehicles" (Participant $\alpha$). This is a particular advantage in rural areas where currently there are "bus service[s] where you've got a 20 or a 50-seater and you're ending up with occupancies of 1 or 2 people" which is "not an economically sensible solution". Participant $\alpha$ goes on to suggest such an automated service would be well suited to on-demand services where "you are reducing the labour cost and the variable costs per mile that aren't fixed". Participant $\beta$ adds that automated services can run for 24 h adding the advantage "that you can work at night [for] better productivity in theory". Participants $\gamma$ and $\varepsilon$ discuss similar advantages but for the business case of delivery services. Demand responsive transport (DRT) solutions such as those described by the interview participants will improve rural community accessibility if implemented effectively [38], particularly at higher levels of automation [39]. Despite the potential to enable greater accessibility (the highest priority), automation remains the lowest ranking rural transport priority. This suggests a perceived disassociation between autonomy and accessibility.

Specifically addressing the consideration of CAEV implementation in rural communities and on rural roads, Figure 1 shows the distribution of responses to the extent of consideration of CAEVs in rural transport planning. In comparison to the consideration of generalised developing transport technologies, consideration of CAEVs is less. Eighty-five percent of participants indicated that CAEVs are "never" or "rarely considered" in rural transport planning.

The flexibility of CAEV DRT to serve rural societies was discussed by four of the five participants. Participant $\gamma$ describes a family situation in which there are multiple destinations for each member to be delivered to and Participant $\delta$ describes less able people with weekly tasks changing each week all being supported by an on-demand CAEV service. Participant $\gamma$ also argues that, despite what some critics of automation might believe, a better connected and automated transport network can generate "footfall in the rural place and actually that's an opportunity for local businesses to ... be connected". Participant $\gamma$ explained that planners are "interested in things like Mobility as a Service, they were interested in car-share, car-pooling, active transport, getting more people on

bikes, more people walking, better ways of connecting between modes of transport that are more sustainable"; however, these may not necessarily involve CAEVs. Participant γ did, however, note that "we'd love to see driverless trains" and that people "still like buses" and CAEV technologies had the potential to "make buses better".

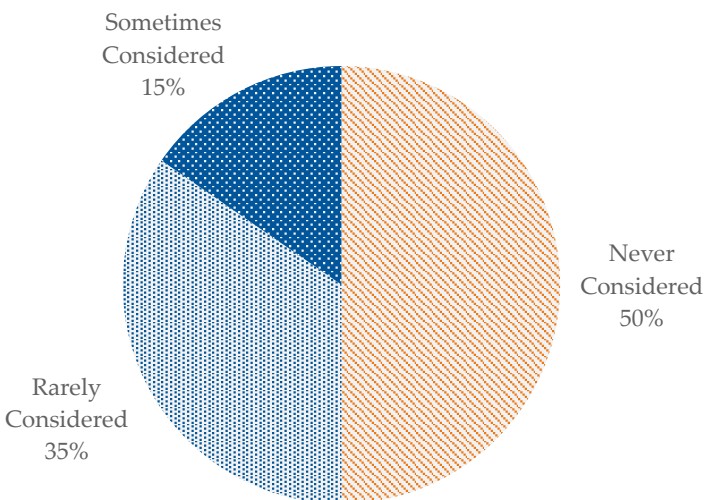

**Figure 1.** Extent of consideration of CAEV implementation in rural transport planning.

Both Participants γ and ε discussed transport hubs in detail and were familiar with the concept. Rural transport hubs are an increasingly popular supporting solution to CAEV and active transport implementation in rural areas which would act to sustain essential rural services including CAEV transport solutions and counteract rural community isolation through physical and digital connectivity [11,65]. Participant γ describes the concept as a "level of the hierarchy of mobility" where rural hubs are connected to urban centres through major transport links (major roads or rail links for example) and connected to their sparse rural communities through smaller, possibly autonomous transport networks. The hubs themselves act as "district centre" with community spaces and activity (Participant ε). Both participants suggested these hubs be used for delivery storage and distribution using CAEVs and unmanned aerial vehicles (UAV) to distribute goods to the surrounding rural region. Participant γ "can see [hubs] springing up more in the countryside" complimented by CAEVs and as co-working spaces.

### 4.3. The Importance of CAEVs in Sustainable Transport Systems for Rural Communities

Transport planners are cautiously optimistic about the benefits of rural CAEV implementation, but do not yet see CAEVs as a priority or an important element of rural transport, at least in the short term. The responses suggest there is great uncertainty about if, and when, this potential will be realised. The interview analysis supports these findings with participants highlighting multiple specifically rural benefits to implementing CAEVs.

The survey asked participants to what extent they agreed CAEVs would improve different aspects of transport serving rural communities. Scores were applied to each response (2 for strongly agree, 1 for agree, 0 for neither agree nor disagree, −1 for disagree and −2 for strongly disagree). The average scores for each aspect are shown in Figure 2.

The results in Figure 2 highlight a tendency for transport planners to "agree" that CAEVs will improve most aspects of rural transportation and a cautiously optimistic consensus for each of the transport aspects, excluding "affordability" for which there is a slight disagreement. This scepticism supports the findings of [13] who identify the affordability of autonomous vehicles as one of the major barrier to rural CAEV implementation. Overall, the cautious results demonstrated in Figure 2 could reflect a lack of understanding amongst transport planners who are aware of the potential but not fully convinced that the technology will deliver.

In Figure 2, "accessibility" and "public transport services" are ranked the most highly. These aspects of rural transport were also ranked as high priorities in Table 1. These results therefore further highlight a potential disconnect between rural community transport needs, the priority of CAEVs in terms of their required technologies and infrastructure (automation, communications, readable roads and charging infrastructure), and the perceived benefits CAEVs could bring.

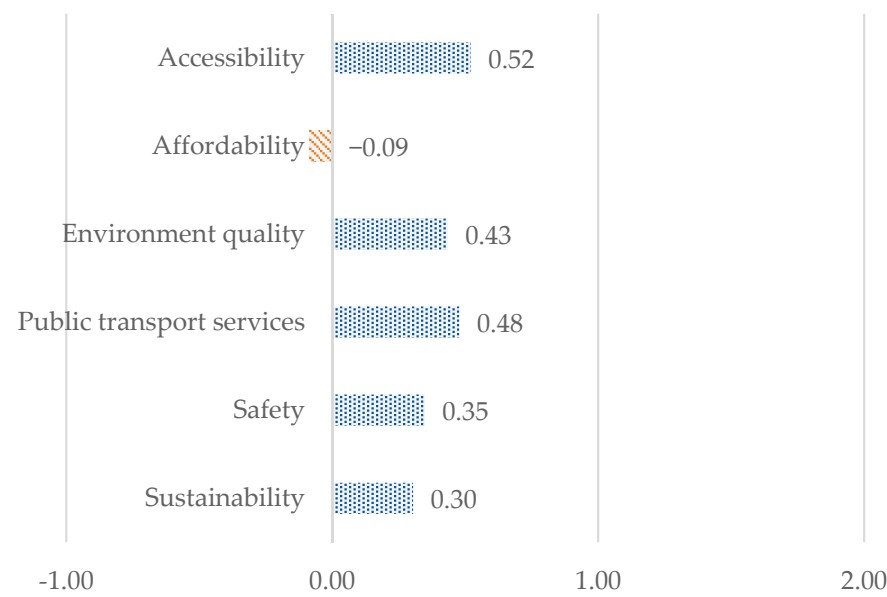

**Figure 2.** The extent to which CAEVs are considered to improve aspects of rural transport.

Survey participants were then asked specifically what advantages CAEVs could bring to rural communities. Through coding of the qualitative responses, Table 2 summarises the foreseen advantages, how often this or a similar advantage was referenced, and gives a summary description of each advantage based on participant's comments.

**Table 2.** Advantages foreseen by transport planners through CAEV technology implementation in rural communities.

| Advantage | Frequency | Summary |
|---|---|---|
| Flexibility | 9 | This advantage was most commonly referenced in relation to public transport services. Connected, autonomous and electric vehicles (CAEV) would provide rural residents with more flexible transport solutions ranging from privately owned CAEV to private or public fleets of taxi, bus or alternative services. |
| Improved infrastructure | 3 | CAEVs could result in less stress on existing road-based infrastructures and provide more opportunities for physical infrastructure improvements and integration for modal shift to alternative transport infrastructure such as rail. |
| Economics | 3 | CAEVs have the potential to reducing the personal cost of transport whilst bringing wider economic benefits through improved connectivity to and between rural communities and urban communities. |
| Ease of planning | 2 | The improved vehicle and system efficiencies of CAEVs could aid route planning and prioritisation. |
| Other | 3 | These included improved transport technologies, access to information, and environmental benefits. |

Despite affordability scoring negatively in Figure 2, economics are referenced several times in Table 2, usually in association with another improvement. This suggests that transport planners are aware of the positive economic potential of CAEVs but in their current state and direction they look to be an expensive solution for the user.

In terms of sustainable transport solutions, CAEVs are overall considered a sustainable solution. However, both the survey and interview results highlight concerns over economic cost, most blatantly highlighted by Figure 2. In the interviews, it was the initial costs of the implementation that were a concern, but high initial costs are to be expected with any new technology and are assumed to reduce as business cases improve with frequency of use. Given the electric aspect of CAEVs, environment quality is expected to improve. Although rarely discussed in the interviews, there was a sense that the benefits of electrification were already well known and therefore did not require explicit discussion. The social pillar of sustainability is that which is expected to benefit the most which was clear across both survey and interview results.

### 4.4. Understanding of CAEV Technologies and Their Rural Implementation Requirements

Wireless communications, well-maintained machine-readable roads and electric charging infrastructure have been identified as critical infrastructures needed to support rural CAEV implementation [13,20,21,23,86].

Based on the survey results machine-readable roads are not considered by transport planners, electric charging infrastructure is rarely considered, and wireless communications are considered only marginally more. Still, this highlights the gap in consideration between machine-readable roads, and electric charging infrastructure and wireless communications. However, although machine-readable roads and their quality is important, on-board CAEV perception technologies, including LiDAR, radar, cameras and the software operating them, are continuing to improve and therefore the extent to which road infrastructure needs to be "machine-readable" is reducing. Whilst electric charging infrastructure and wireless communications infrastructure are also required for CAEVs to operate successfully, these technologies also feature in many currently active personal road vehicles. For example, EVs currently on the roads require charging infrastructure, and many feature vehicle software that receive over-the-air updates. As such, these two infrastructures are likely to be a higher priority for rural transport planners to establish. Despite this, both remain rarely considered.

Survey participants were also asked to rank what they perceived to be barriers to rural CAEV implementation, and therefore what needs to be addressed as a requirement for implementation. Firstly, "government acceptance" stood alone as the front-running barrier to CAEV implementation. "Communications", "regulation", "public acceptance" and "electric charging infrastructure" were ranked next, all scoring similar averages (Table 3). "Industry acceptance" and "machine-readable roads" were seen as the least significant requirements for CAEV implementation.

**Table 3.** Ranked barriers to rural CAEV implementation.

| Barrier | Rank | Score | Range |
|---|---|---|---|
| Government/local authority acceptance | 1 | 4.9 | 5.0 |
| Communications infrastructure | 2 | 4.4 | 5.0 |
| Regulation and law | 3 | 4.3 | 6.0 |
| Public acceptance | 4 | 4.2 | 6.0 |
| Electric charging infrastructure | 5 | 4.1 | 6.0 |
| Industry acceptance | 6 | 3.1 | 6.0 |
| Machine-readable roads | 7 | 3.0 | 5.0 |

Despite these average rankings, the range of responses was dramatic with each of the seven barriers being ranked from first to sixth by at least one participant in each case. This highlights a lack of consensus within the rural transport planning sector on the issue of barriers to CAEV implementation. There could be several reasons behind this finding, including a lack of universal understanding of these barriers, difference in opinion based on local circumstance, or that all the barriers listed are significant based on the different perspectives of individual respondents.

When discussing technological challenges of rural CAEV implementation with the interview participants, there were two main areas of discussion. Firstly, on-board technologies including perception, communication and software within the vehicle are required to operate in the "extremely challenging" rural environment. Participant α suggests there is a need for rural CAEV developers "to concentrate solely on what's available on the vehicle" with Participant δ supporting this citing that CAEV companies "would rather the vehicles were as self-contained as possible". This would mean that rural CAEVs would be as reliant as possible on their on-board technologies, rather than the rural infrastructure around them. Keeping operations internal helps to avoid "cybersecurity issues" (Participant α, Participant δ) and reduces the need to upgrade or install infrastructure in rural areas which is widely viewed as a challenge. In terms of high-level automation such as those of SAE Levels 4 and 5, Participant α thinks "there's still a lot of uncertainty around when you're actually going to see those levels of automation" particularly in challenging rural environments. "Jumping to level five [CAEV technology] . . . has to be a significant time away from now. If you look back at the predictions that were made five years ago, we haven't made five years of progress at all". Secondly, come the technological challenges of CAEV testing in rural areas. Participant α refers to simulation and test-track trials which are "gradually solving problems"; however, they don't think "anyone has really worked out how to do the [real-world] testing properly". Participant δ believes that real-world testing is justifiable, particularly in the rural case, to "learn about edge-cases so that they can go into programming and minimise the likelihood of problems further down the line", where 'edge-cases' refer to challenging "situations that are rare but will cause a lot of problems for an autonomous vehicle". Participant γ explains that rural policy makers are beginning to see the potential of rural CAEV testing but they are not getting anyone "knocking on their door [asking if they] can run a pilot", although they believe that "they probably should be because it's probably a safe place to do a trial if you can do it on a small scale". Despite the current transport dangers on rural roads, Participant β notes that "the great thing about rural environments . . . is that there aren't dense populations so in a way it is great from a safety point of view" and goes on to describe the opportunities for delivery applications such as drone and pavement-sharing vehicles which could be dangerous in busy cities. Participant ε identifies an opportunity for CAEVs to make use of rural off-road trails and public rights of way which they describe as "fantastic routes that have got thousands of years of history of connectivity and they were made to connect these villages up together in the straightest route that you could". Participant γ believes that the "interface between the technology, the investors and innovators and the rural [is] the biggest barrier" to rural CAEV implementation and is defined by the level of testing. Explaining the lack of interest in trialling rural public transport services, Participant α describes a "vicious cycle" in which individuals living in "patchy, underserved" rural areas are forced to "invest in [their own] private car" which further decreases demand for already limited public transport services.

### 4.5. Elicitation Results Implications and Proposed Solutions

Ninety-six percent of survey participants noted that any understanding of CAEV technologies amongst the transport planning industry was either rare or entirely absent. Most of these were the same respondents reacting similarly in Figure 1 believing that CAEVs were not or rarely considered amongst transport planning professionals as a rural transport solution. This lack of understanding is supported by the previous findings in this results section of a perceived disassociation between CAEV technologies and rural

transport priorities, and the lack of consensus amongst the transport planning profession regarding rural CAEV implementation challenges.

Conducting the interviews with selected experienced transport professionals enabled further investigation into transport planner understanding of CAEV technologies and their rural implementation requirements. Participant α mentioned that there was no "guidance available" for local planning authorities on CAEV implementation. This related to the uncertainty around CAEV technologies and technological progress as well as timescales of when and if this technology will be available. Participant β suggested that CAEV implementation would be "based on local decisions" as there is currently no "form of government politicisation" on the direction of CAEVs, their technologies and energy infrastructure. Participant γ echoed this and explained that "there is so much diversity of need [in rural areas] it's down to local authorities" to assess the requirement, identify barriers and opportunities and prioritise areas of investment. However, "the danger with that is you get a more fragmented delivery of technology" and Participant δ notes that there are "too many different county councils all doing their own thing". Participant ε further explained that district highway authorities can suggest technical specifications for local planning authorities but "to get [local authorities] to all work together may be problematic because they are very localised . . . , very traditional [and] most of them are conservative", which is a particular problem when highway authorities cover both urban and rural areas. Further, Participant α acknowledges that local authorities consist of "very small teams with a huge variety of responsibilities" and these responsibilities vary depending on local circumstance. In terms of progress towards CAEV implementation or even consideration, Participant α suggested that "it depends how close they've been to the technology" and that "some of the councils are extremely well informed and in some cases, they are actually active participants" in CAEV trails. However, Participant α did note that most of the councils that they had observed who are engaged in CAEV trials were urban-based planning councils.

These comments highlight a lack of national direction of CAEV implementation resulting in fragmented understanding of CAEV requirements and potential. These findings reflect those of [8] who demonstrate that governance and legislation are strong barriers to the implementation of alternative transport services such as CAEVs. Further, [13] identifies the politicisation of transport planning as a barrier to rural CAEV adoption, noting that policy implementation often fails rural areas in favour of urban. There is a uniquely rural challenge here in that the rural communities that CAEVs could potentially serve are geographically, socially and economically diverse. This suggests that it will be difficult for national policy to deliver a broad solution that solves the unique rural transport challenges through CAEV implementation facing rural local authorities.

Whilst the lack of understanding is clear and challenges abundant, Table 4 summarises the suggestions of survey participants as to how transport planners could better understand CAEV technologies and their implementation requirements. The suggestions were analysed by coding similar qualitative ideas into group nodes. The frequency column in Table 4 indicates how often that idea, or equivalent ideas were suggested. The frequency of the ideas exceeds the total number of respondents due to some respondents offering multiple suggestions, but also due to overlapping suggestions such as proof of technology.

Participant ε believes that a combination of solutions such as those in Table 4 would be effective in getting the CAEV implementation processes underway. They identify that highway and transport planners in particular are "already on board [with CAEV implementation]" because they are "already thinking of 20 or 30 years ahead" as "that's how they operate". However, local "planning authorities are often a bit slower" in this case to realise the technological potential of CAEVs. Engagement with and between highway and local authorities on the matter of CAEV implementation is an important step which should be combined with specific rural CAEV case studies as proof to inform planners on CAEV potential and implementation requirements.

**Table 4.** Advantages foreseen by transport planners through CAEV technology implementation in rural communities.

| Advantage | Frequency | Summary |
| --- | --- | --- |
| Stakeholder engagement | 7 | Engagement between planners and stakeholders involved specifically in CAEV and technological development to encourage knowledge sharing and spread awareness. |
| Proof of technology | 6 | Experimentation and demonstration to prove that CAEV and related technologies actually work, ideally in real-world conditions. Proof of safety and a range of benefits. |
| Formal education | 6 | Traditional education methods such as CPD and training but also including written forms of communication such as formal guidelines for best practice. |
| Case studies | 5 | Completed case studies showing proof of implementation in specific scenarios, can either be in written form or demonstrated first-hand. |
| Economic investment | 5 | Economic investment in CAEV trails and projects helps to raise awareness and understanding, particularly large and high-profile investments. |
| Policy change | 4 | Formal changes to policy and legislation in effect force planners to acknowledge and understand the requirements for CAEVs. |
| Other | 2 | Physical and interactive modelling; generic knowledge sharing. |

Whilst engagement between planning authorities is important, so too is cross-engagement with the industries and researchers developing CAEV technologies. With industry, rural CAEV implementation primarily depends on business case, markets and profitability with economics being a key challenge group discussed by the interview participants. Industry needs "continuous improvement in technology and by making [CAEV technologies] more accessible and affordable you're improving your economic case and business case" (Participant β). To help achieve this, Participant β then suggests that government, transport and local authorities need to work with industry and subsidise them until the balanced is reached. The challenge for government is around certification to allow industry and planning authorities to implement CAEVs and their technologies. Participant α explains that there are government certification agencies, of which "the DfT is part of informing that process", that "see systems go through an approval process [and] certification process". Despite this, Participant α believes that "a lot of those processes don't really exist" for CAEV technology. This means that "these certification agencies need time to [develop] a different set of skills within them to actually do the certification of a very different system" to what they are used to.

Based on the findings of this paper, before CAEVs can be implemented in rural communities and areas, better understanding of the technologies and requirements is needed. Yet to develop this understanding, better inter-institutional communication is needed to consolidate the rural transport challenges that need solving (inclusive of rural mobility dynamics [15]), the potential of CAEV technologies to solve those challenges and the technological and infrastructural requirements of CAEV implementation within specific rural scenarios. Participant β believes that the UK suffers from these institutional communication and engagement challenges, particularly in terms of transport technologies. "Compared to other industrialised nations, as a country we suffer from technology translation." Participant β cites the German Fraunhofer Society as an example which consists of "large industrialisation centres" which bridge the gap between university research and industrial manufacturing. "One thing that the government can do to accelerate [the development and

implementation of CAEVs and their technologies] is create a translation centre to cover some of that ground [of technology translation]."

## 5. Conclusions

CAEVs have the potential to provide sustainable transport services, bring societal and economic benefits, and solve many of the specific transport challenges facing rural communities. The challenges facing CAEV implementation include the effectiveness of on-board technologies operating in and integrating with rural road environments and infrastructure; but they also include a lack of understanding of CAEV technologies as identified by 96% of surveyed transport planners.

The depth and breadth of the challenges to and potential for rural CAEV implementation gained from the elicitations adds to this paper's understanding of the issue. Whilst transport planners recognise the major rural transport needs as identified in this paper, they do not necessarily identify CAEVs as a potential solution unless prompted. This it reiterated by the identified lack of technological transport systems in rural communities and 85% of rural transport planners rarely or never considering CAEVs as specifically rural solutions. Despite this, transport planners recognise that CAEV technologies may be able to provide some of the benefits highlighted, but a greater awareness and understanding is needed before progress in this area can be made. The findings of this paper echo those of [72] where transport planners have generally positive attitudes towards emerging transport technologies but do not have the capacity to effectively implement them.

There are options to counter this lack of understanding, and the range of suggestions from survey participants imply a willingness for transport planners to engage with CAEV awareness and education, dependent on factors including relevance and time. This was echoed by the interview participants who could see the potential but were unsure of the next steps. Despite this, the challenges identified by transport planners themselves centre on the understanding and acceptance of CAEV technologies by government and local authorities. Further, there is a need for inter-institutional communication and engagement to develop CAEV understanding and implementation strategies for which several options have been identified. This conclusion is similar to that of [11,13], who go further to stress the importance of including rural communities in transport planning decision making, which is particularly essential in the rural context.

Through elicitation exercises with transport planners, this paper has identified the extent to which CAEV technologies are currently a consideration and priority for rural transport planning professionals implementing sustainable transport solutions for rural communities. This paper has also begun to identify the extent to which transport planning professionals, and wider institutions, understand CAEV technologies and their infrastructural requirements. In addition, several other requirements needed to aid rural CAEV implementation have been identified, the most conspicuous being the engagement with and between institutions developing, regulating and implementing transport technologies.

This paper hypothesised that a lack of specifically rural implementation research and trials meant that transport planners were likely to be ill-informed and uncertain of both the potential of CAEVs and their implementation requirements. To an extent, this hypothesis was found to be true where the understanding of CAEV potential to alleviate rural transport challenges, notably accessibility, was lacking and the challenges facing implementation diverse and non-specific.

Urban-based CAEV implementation was also a dominant theme throughout this research. Based on this and the identified need for effective institutional engagement and collaboration, this paper supports the finding of [11] who identify the need for a future rural mobility strategy supported by digital and transport technologies to facilitate rural development. These findings challenge the Department for Transports Future of Mobility Strategy which assumes that CAEV successes in cities can be transferred to rural areas [87]. This strategy overlooks the innovation potential of specifically rural CAEV implementation based on the specific needs of rural communities.

## 6. Further Work and Recommendations

Whilst this paper is not a thorough investigation of UK transport planner perspectives on the issue of rural CAEV implementation in the UK, it does address the unique challenges facing rural implementation of CAEV technologies, hard and soft supporting infrastructure and stakeholder engagement and understanding explicitly, for the first time. Transport planners are essential to the implementation of widespread CAEV technology and as such, need to be better informed and engaged with both academic and industry-based research. This paper contributes to acknowledging this gap and acts as a platform from which to build upon these findings.

The authors suggest further work is needed to substantiate the results of this research study in the UK. Specifically, a greater survey and interview sample size in needed. Further research could extend to other countries where CAEV technologies are reaching implementation stages. In addition, action must be taken to ensure that rural communities do not lose out as they have done historically to urban bias in transport and technological development. This can be achieved through engagement with transport planning bodies, professionals and the public together with the development of methods to aid understanding and improve access to these technologies and their requirements. Such methods, of which some are outlined in this paper, should encourage rural-based CAEV trials and implementation to enable effective technological rollout that specifically benefits and connects the rural communities they are intended to serve.

**Author Contributions:** Conceptualization, J.G.W.; Methodology, J.G.W.; Validation, J.G.W., S.M. and L.R.; Formal Analysis, J.G.W.; Investigation, J.G.W.; Data Curation, J.G.W.; Writing—Original Draft Preparation, J.G.W.; Writing—Review & Editing, J.G.W., S.M. and L.R.; Visualization, J.G.W.; Supervision, S.M. and L.R.; Project Administration, J.G.W.; Funding Acquisition, J.G.W., S.M. and L.R. All authors read and agreed to the published version of the manuscript.

**Funding:** The primary author is in receipt of a fully funded EPSRC Doctoral Training Grant studentship (grant number: EP/R513283/1).

**Data Availability Statement:** The data presented in this study are available on request from the corresponding author. The data are not yet publicly available as the primary author's PhD thesis has not yet been formally submitted. Further, the identities of this paper's research participants must be protected.

**Acknowledgments:** The primary author would like to acknowledge the financial support from the EPSRC which has enabled this study. Thanks also go to each of the survey and interview participants for committing their time and insights to this research. The author also acknowledges the Nottingham Geospatial Institute and Transport, Mobility and Cities Group at the University of Nottingham, Transport Planning Society, and the Royal Institute of Navigation for their sup-porting roles in completing this research. Final thanks go to the primary author's supervisors Stuart Marsh and Lucelia Rodrigues for their support and contributions to this paper.

**Conflicts of Interest:** The authors declare no conflict of interest.

## Appendix A

Survey Question Schedule

1. In which country are you based?
2. In which region or city is your work based?
3. Who is your employer?
4. Briefly describe your job role.
5. Please rank the following priority areas for rural transport in your region, with the highest priority first. Accessibility; affordability; automation; communications infrastructure; environment quality; maintenance of infrastructure; public transport services; quality of infrastructure; safety; sustainability.
6. To what extent do you agree that urban transport planning takes priority over rural transport planning?

7. To what extent do you agree that future transport systems and technologies are considered when planning rural transport systems and infrastructure?
8. Are CAEVs considered in rural transport planning in your region?
9. To what extent are the following CAEV supporting infrastructures considered in rural transport planning? Electric charging infrastructure; machine-readable road features, markings, and signage; wireless communication networks.
10. Please rank the following barriers to rural CAEV implementation, with the largest barrier to implementation first. Communications infrastructure; electric charging infrastructure; industry acceptance; machine-readable road features, marking and signage; public acceptance; regulation and law.
11. To what extent do you agree that CAEVs will improve the following aspects of rural transport? Accessibility; affordability; environment quality; public transport services; safety; sustainability.
12. Please state any other areas of rural transport that you believe CAEVs will improve.
13. In your opinion, how well are CAEVs, their technologies, and their planning requirements understood amongst the rural transport planning industry?
14. Please suggest how the understanding of CAEV planning requirements could be improved?

Interview Question Schedule

1. Can you tell me your job title and explain your job role?
2. Where is your work based or what geographic regions does your work cover?
3. When thinking about rural transportation what are the priority areas, or problems that need to be addressed?
4. There is an urban focus for CAEV trials, do you think this is justified and why? Does this bias impact other aspects of transport?
5. Do transport planners consider CAEVs and their infrastructure requirements when developing rural transport solutions? To what extent they are CAEVs considered, and in what scenarios?
6. What are the benefits of implementing CAEVs on rural roads? How can these benefits be realised?
7. Are there any barriers to rural CAEV implementation? Are there strategies to overcome these?
8. To what extent do transport planners understand CAEVs and their technologies? What can, or is anything being done, to improve understanding?

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
