# Peer review of "Planning Perspectives on Rural Connected, Autonomous and Electric Vehicle Implementation"

_sustainability, doi:10.3390/su14031477_

Round 1
Reviewer 1 Report
Answers for Authors
Dear Authors, thank you very much for submitting your paper " Planning Perspectives on Rural Connected, Autonomous and Electric Vehicle Implementation"
After reviewing heavily reviewing your paper i came to the following conclusions: The paper need to be revised.
The paper is overall of good merit but lacks some information, state of the art literature, structural refinements and some additional explanations. In the following i will display what needs to be added to the paper:
1. Overall Feedback:
- this is a very well written paper with an good overall research design and some interesting findings.
2. Abstract, Introduction and General Objective:
- Abstract: very well written and includes everything
- Introduction: very well written and includes everything.
- General Objectives/Main contributions: I am missing a list of clear contributions. Altough you pointed out your hypothesis, it would be good to have a clear list of contributions e.g. 1., 2., 3. at the end of your hypothesis
3. State of the Art and Literature used: This is a very broad state of the art which you displayed here. I like all the papers you integrated and i think this state of the art is almost complete. For me The state of the art is missing papers in the field of mobility planning, spatial planning, and mobility movement estimation. This kind of research is displaying what models you can use to extract mobility behavior especially in rural areas too. I think you should have an additional look at this paper and integrate them:
- Filippo Simini et al. "A universal model for mobility and migration patterns" Nature vol. 484 no. 7392 pp. 96-100 2012.
- Hugo Barbosa et al. "Human mobility: Models and applications" Physics Reports vol. 734 pp. 1-74 2018 ISSN 03701573.
- D. Ziegler, J. Betz, and M. Lienkamp, “Unified Mobility Estimation Model,” 2021 IEEE International Intelligent Transportation Systems Conference (ITSC). IEEE, Sep. 19, 2021. doi: 10.1109/itsc48978.2021.9564453.
One paper that is covering this topic similarly is also missing and that is the paper of "D. Prioleau, P. Dames, K. Alikhademi, and J. E. Gilbert, “Barriers to the Adoption of Autonomous Vehicles in Rural Communities,” 2020 IEEE International Symposium on Technology and Society (ISTAS). IEEE, Nov. 12, 2020. doi:´0.1109/istas50296.2020.9462192." Check out his additional references too. You can learn something from this research, too
4. Methods/Algorithms/Approaches:
- You said you are combining surves and semi-structured conversation interviews. I am issing the survey questions and interview questions in this paper. Can you add them to the appendix?
- What is for you a rural area? This needs to be defined explicitly in the context of this paper.
5. Results:
- "In total, 23 survey responses from professionals in planning and planning-related disciplines (referred to as survey participants) were analysed supported by five detailed interview elicitations" - How are you justifiying this small number of feedback for a survey? is this really enough? Is this really displaying a broad background?
6. Discussion: You are missing your discussion section completely. From a scientific paper point of view you need to enter a critical and reflective discussion. Explain to the reader what is good on your approach and what is bad. Be really honest here what needs to be done in addition to enhance your approach further. In addition you can draw here conclusions to other authors.
In addition i think to make the paper more complete it would be helpful for everyone if you open-source your code on Github or similar.
Author Response
Thank you for your detailed review and comments. Based on your feedback we have amended our paper as follows:
- A set of research contributions has been added at the end of the hypothesis.
-
We have reviewed the suggested literature and written a new paragraph in the CAEV section to reflect the key themes you suggested. This also supports the papers exploration of the potential for rural CAEV implementation. The literature has also been integrated throughout the paper where relevant, including a contribution to the papers discussion section, supporting the conclusion for better inter-institutional communication and teamwork.
-
We have integrated contributions from the Prioleau paper suggested and its own citations throughout the paper, both to support statements already made and to contribute additional challenges relevant to the scope of the paper.
-
We have defined "rural" in the methodology section.
-
We have included the survey and interview question schedules in an appendix.
-
We have expanded the results sections to include a broader discussion regarding of the small number of responses which is more explicitly referenced in the conclusions and further work section.
-
We integrated the discussion section with the results because we felt this created a better sense of flow for the reader and replicated the conversational style of the interviews. We have renamed this section "Elicitation Results and Discussion“. We have also integrated the conclusions of other authors into this section, in particular highlighting the findings of Prioleau who’s work you suggested.
-
We have included the survey and interview question schedules in an appendix. There is no code to make available and any other relevant results would threaten the anonymity of the participants if made available.
Reviewer 2 Report
Very important topic. Authors prepared intersting survey. They explaided in details why they used selected approach to get sample.Literature review is enough in my opinion.
Of course we should use the paper as preliminary results - I mean as a begining of the next step of survey. In the case in my opinion the paper is good and ready to publish.
In future Authors should take into account to add more Figures. Graphics may turn paper to more "readers-friendly" level. However current version is also good.
Author Response
Thank you for your positive review and comments. We have further expressed the preliminary nature of the results and suggested further studies with greater sample sizes. We have also added the survey and interview question schedules as an appendix.
Reviewer 3 Report
The manuscript has some grammatical errors and typos. It is necessary to write the acronyms in extended form when they appear for the first time in the text It is necessary to extend the literature review in the introductory paragraph. For example, it is suggested to consider the development of AVs in the light of smart cities and the criticalities imposed by the recent pandemic. Therefore, we recommend reading the following research works 1)Severino, A., Curto, S., Barberi, S., Arena, F., & Pau, G. (2021). Autonomous Vehicles: An Analysis Both on Their Distinctiveness and the Potential Impact on Urban Transport Systems. Applied Sciences, 11(8), 3604. 2) Campisi, T., Severino, A., Al-Rashid, M. A., & Pau, G. (2021). The development of the smart cities in the connected and autonomous vehicles (CAVs) era: From mobility patterns to scaling in cities. Infrastructures, 6(7), 100. 3)Liu, T., hai Liao, Q., Gan, L., Ma, F., Cheng, J., Xie, X., ... & Liu, M. (2021). The role of the hercules autonomous vehicle during the covid-19 pandemic: An autonomous logistic vehicle for contactless goods transportation. IEEE Robotics & Automation Magazine, 28(1), 48-58. 4)Toapanta, A., Zea, D., Tasiguano, C., Anangano, G., Prado, A., & Camacho, O. (2021). A review of autonomous vehicle technology and its use for the COVID-19 contingency. Ciencia E Ingenieria, 43-51. A flow chart / graph in the introductory part of the manuscript could help in understanding the research and related stepsit is necessary to insert a more detailed description of the case study and of the implementation and administration of the questionnaire/survey
Author Response
Thank you for your detailed review and comments. Based on your feedback we have amended the paper as follows:
-
Grammatical errors and typos reviewed and corrected.
-
Acronyms reviewed and extended where missing.
-
We have integrated the suggested literature where appropriate to support the paper's literature review and findings, thank you for drawing our attention to these. In particular, the paper has now expanded on the potential influence of the pandemic on the future of CAEV implementation and rural mobility.
-
We have considered the suggestion of a flow chart but as per comments from the other reviewers we do not believe this is necessary on this occasion.
-
We have included the survey and interview question schedules in an appendix and made reference to this within the paper text.
Round 2
Reviewer 3 Report
the manuscript has some grammatical and formatting errors; once these have been corrected it will be eligible for publication
Author Response
Thank you for your further feedback. We have corrected the spelling and grammatical errors. We have also rearranged the tables and figures to each fit on their own page and formatted the references section using the correct style.